# GlotCC: An Open Broad-Coverage CommonCrawl Corpus and Pipeline for Minority Languages

**Amir Hossein Kargaran**♣    **François Yvon**♠    **Hinrich Schütze**♣

♣LMU Munich & Munich Center for Machine Learning, Munich, Germany
♠Sorbonne Université & CNRS, ISIR, Paris, France
amir@cis.lmu.de

## Abstract

The need for large text corpora has increased with the advent of pretrained language models and, in particular, the discovery of scaling laws for these models. Most available corpora have sufficient data only for languages with large dominant communities. However, there is no corpus available that (i) covers a wide range of minority languages; (ii) is generated by an open-source reproducible pipeline; and (iii) is rigorously cleaned from noise, making it trustworthy to use. We present GlotCC, a clean, document-level, 2TB general domain corpus derived from CommonCrawl, covering more than 1000 languages. We make GlotCC and the system used to generate it— including the pipeline, language identification model, and filters—available to the research community.

🤗   **Corpus**   v.1.0   hf.co/datasets/cis-lmu/GlotCC-v1
⭘   **Pipeline**   v.3.0   github.com/cisnlp/GlotCC

## 1 Introduction

Progress in building language technologies has largely been limited to about 200 languages (referred to as "large languages" in this paper) for which text is reasonably accessible [34]. In this regard there is a disparity between large languages and minority languages, a disparity that becomes much more visible as state-of-the-art language technologies – developing from word embedddings and encoder-only models to large autoregressive models – become more data-hungry. FastText [14, 28] word vectors supports ≈160 languages, XLM-R [22] ≈100 languages, Llama-3 [51] ≈30 languages. So there is a trend for the number of supported languages to decrease over time in state-of-the-art models.

While efforts have been made to compile multilingual corpora from books [45, 26], the most common approach to collecting large amounts of raw textual data is to rely on crawled web text [55, 4, 11, 42, 67]. However, there are still many issues to resolve to guarantee the quality of the extracted web text, particularly for minority languages [41]. Most pipelines for web-text extraction rely on language identification (LID) to classify text, e.g., on FastText [14, 35] (as does OSCAR [4]) or on CLD3 [15, 64, 73]. Many errors in existing web corpora are artifacts of their LIDs [20]. For example, FastText's accuracy suffers from hash collisions: it maps out-of-domain (OOD) n-grams to n-grams seen in training, resulting in many OOD errors. Additionally, commonly used LIDs can identify the (roughly) 200 largest languages but only a few minority languages, leading to "out-of-model cousin" errors [20, 41].

In this paper, we adopt the Ungoliant pipeline [3] for extracting text from CommonCrawl. To address the limitations of current LID models (hash collisions and limited language coverage), we develop a new LID model, GlotLID v3.0, an extension of GlotLID v1.0 [37]. This model covers over 2,000

linguistic labels. It also mitigates common sources of noise encountered on the web with a better rejection model. We also extend Ungoliant with several filtering techniques that remove general web noise, list-like content, documents with repeated words [42, 62] and "inconsistent" documents, i.e., documents for which LID detects multiple languages, often an indicator of noise. We perform an audit of random 653 language subcorpora of GlotCC, finding that the data is in-language, with a macro-average score of 0.93 and median score of 1.0. We publish GlotCC as a document-level corpus. This makes it usable for pretraining generative language models as well as for other language technologies that require information beyond the sentence level.

## 2    GlotLID

In our previous work, we introduced GlotLID v1.0 [37], an LID model based on the FastText architecture [14] with good performance for 1665 languages. Upon further analysis, we improved GlotLID to v2.0 and then v3.0. In v2.0, we extended our ISO 639-3 labels to also include the ISO-15924 script; an example is rus-Cyrl (Russian language in Cyrillic script). This new labeling reduces errors and enhances the quality of language resources. For instance, it allows us to restrict the set of language labels that can be assigned to each script, thereby avoiding obvious mismatches between language and script [40]. This can be implemented by employing writing system detection tools, such as GlotScript-T [40], during inference. We also use GlotScript-T and GlotScript-R to filter out noisy training data from the GlotLID training corpus. GlotScript-T determines the script of an input text. GlotScript-R provides the admissible script(s) for each language.

In GlotLID v3.0, we increase the number of LID labels to more than 2000 by adding, removing and relabeling resources based on feedback from the community and on our own findings from the continuous analysis we are conducting.

### 2.1    GlotLID v3.0 vs GlotLID v1.0/v2.0

#### 2.1.1    Increased coverage of minority languages

We include in the GlotLID training set new resources for African languages [43, 66, 7, 56], Uralic languages [31, 74], Indonesian languages [70], Indic languages [48, 25] as well as additional indigenous languages [19]. We also consider data for a variety of languages from books [45], children storybooks [2], crowdsourcing [10], low-resource crawls [1], and world language repositories [54].

#### 2.1.2    Better rejection model with the "UND" label

LID is typically understood as a closed-set classification problem; most LIDs, including GlotLID, adopt this setup. As LID is in fact an open-set problem [32, 49], when processing web data, there is always the risk of encountering "unknown" languages (those not occurring in train). To limit errors that are caused by these unknowns, it is customary to reject samples for which the most likely language has a low probability, assuming the classification model is well-calibrated [68, 37].

This problem is compounded when using popular LID architectures such as FastText and CLD3, which detect languages based on character n-gram features. To limit memory footprint and speed up inference, FastText hashes character n-grams into a predefined range of integers, using this map to retrieve feature embeddings. This procedure applies to utf-8 character $n$-grams, where the default for GlotLID is $2 \leq n \leq 5$.

This approach makes no distinction between n-grams that have been seen in training and those that have not, as any n-gram is assigned to an existing hash value. Furthermore, as the number of languages increases, so does the number of scripts, and accordingly, the number of possible n-grams. With 160+ scripts, this number is in fact much greater than FastText's default hash size (2,000,000), increasing the chance of collisions. This first implies that the probability of languages that are well represented in the training data (i.e., high-resource languages) or that have a large number of n-grams due to their character set (e.g., Chinese) is overestimated. This also means that closed-set LIDs will provide a non-zero probability score even for writing systems that were never seen during training, predicting (sometimes with high probability) languages whose n-grams collide with unseen n-grams.

Although GlotLID supports all major scripts, its training data does not contain minor scripts such as Mong (Mongolian), Sylo (Syloti Nagri), Newa (Pracalit), Talu (New Tai Lue) and Gran (Grantha). In

order to correctly reject languages written in these scripts, and given that we have no access to reliable training data for them, we introduce a set of 157 new "und" (*undetermined*) labels. The associated training data is obtained by randomly generating character strings from the corresponding character set. For example, for the Talu script, we use the label "und_Talu". "und_Talu" training data consists of randomly selected Talu characters forming 100,000 sentences.

### 2.1.3 Removing noise with "zxx" labels

In addition to handling unseen scripts, web crawlers and processing systems also need to be robust to multiple types of noise [20]. To make GlotLID more robust, we create training data for additional types of noise, including mis-rendered PDFs and Mojibake (text decoded using an unintended character encoding). We either create artificial training data or we use an initial seed and query the Google search engine to obtain training data.

We include six major sources of web noise that we encountered in our work on GlotLID.

1) **Mis-rendered PDF:** This is gibberish Latin text that appears when an Arabic PDF is mis-rendered by OCR. To generate representative sentences, we query 1- and 2-grams of the letters 'i', 'j' and 'l', with spaces between them, as these letters are frequent in this context.[1]

2) **A N T S P E A K:** This is a type of noise where the characters of the text are space-separated [20]. It is easy to generate synthetic training data from any text or alphabet.

3) **Binary files:** This type of noise occurs when binary files, especially images, become part of the text. It is easy to generate training data by opening any binary file in a text editor.

4) **Mojibake Latin:** This is gibberish text that results from text being decoded using an unintended Latin character encoding. In this type of noise, vowel characters of latin with different accents in a repeated form can serve as a seed for Google queries, such as "áàãà".[2]

5) **Mojibake Arabic:** This is gibberish text that results from text being decoded using an unintended Arabic character encoding. In this type of noise, ^, ®, ±, § and the Arabic characters Tah and Zah appear frequently. Our queries are based on these characters.[3]

6) **Replacement character:** The replacement character (U+FFFD) appears repeatedly in this type of text. We simulate it by randomly replacing characters with the replacement character.

We create three GlotLID labels for these six noise types: zxx_Latn for noise types [1-4], zxx_Arab for noise type 5 and zxx_Zzzz for noise type 6.

### 2.1.4 Curation of labels

To curate our set of LID labels, we rely on confusion matrices, focusing on languages with low performance, i.e., those that are frequently confused with others. We employ both genealogical analysis and basic sanity checking using web resources about the languages. See Section 7 of [37] for an example of our methodology. Most of the problems identified this way were due to the training data, e.g., the training corpus for a given language is actually a mix of several languages. We remove labels and their training data if we deem them too noisy based on our analysis.

GlotLID v1.0 and v2.0 both support macro languages and individual languages. In cases where the correct class is an individual language, the associated probability is often spread over this language and its macro language. The reason is that many individual language n-grams and words are also frequent in the macro language. In GlotLID v1.0/v2.0, we provide the option of taking the softmax on a subset of LID labels (e.g., on just the individual language and the macro language). However, according to community feedback, it is preferable for labels to be mutually exclusive, making it possible to run LID just once over all supported labels. We rectify this problem by (i) re-labeling macro languages as one of the individual languages; (ii) merging the individual languages into the macro language (in case of small differences between them, e.g., we merge individual languages prs_Arab and pes_Arab into macro language fas_Arab); and (iii) deleting the macro language in case we already have good support for its individual languages such as "zho_Hani", "aze_Latn", "est_Latn". However, there are

---

[1]Mis-rendered PDF query example: `google.com/search?q=i+j+l+ii+jj+ll+ij+ji+lj+jl+li+il`
[2]Mojibake Latin query example: `google.com/search?q=Ãą Ãã Ãč Ãã`
[3]Mojibake Arabic webpage example: `al-jazirah.com/2010/20100802/index.htm`

some conventional exceptions: we keep both "srd_Latn" and "sdc_Latn". "srd_Latn" only represents "sro_Latn" and "src_Latn", not "sdc_Latn". This decision is based on the Glottolog tree [29] and the linguistic map of Sardinia. We changed "tgl_Latn" to "fil_Latn" following Kudugunta et al. [42].

## 2.2 Evaluation setup

We train GlotLID v3.0 using the same parameters and sampling strategy as GlotLID v1.0 [37] (also described in Table 1) for 1 epoch. In our previous work [37], we found that the variance of the FastText architecture with different initial seeds is negligible.

Table 1: GlotLID v3.0 training hyperparameters

| Argument | Description | Value |
|---|---|---|
| -minCount | Minimal number of word occurrences | 1000 |
| -minCountLabel | Minimal number of label occurrences | 0 |
| -wordNgrams | Max length of word ngram | 1 |
| -bucket | Number of buckets | $10^6$ |
| -minn | Min length of char ngram | 2 |
| -maxn | Max length of char ngram | 5 |
| -loss | Loss function | softmax |
| -dim | Size of word vectors | 256 |
| -epoch | Number of epochs | 1 |
| -lr | Learning rate | .8 |

GlotLID is trained on the GlotLID v3.0 corpus, which draws from many different sources; the three main sources are Wikipedia, news websites, and religious texts.[4] The new languages (§2.1.1), und labels (§2.1.2), and zxx labels (§2.1.3) are also included. As some of our evaluation benchmarks (§2.2.1) might have leaked into other sources included in the GlotLID corpus, such as UDHR in articles (e.g., Wikipedia), translation community resources (e.g., Tatoeba), and news (e.g., BBC), we remove this contamination from the GlotLID corpus. We count a benchmark test sentence as occurring in the GlotLID corpus if all of its word four-grams occur in one sentence of the GlotLID corpus. We remove all of the sentences from the GlotLID corpus that meet this condition. After deduplication, we split the corpus into training, validation, and test sets in the ratio 0.8/0.1/0.1.

Following prior work [55, 37, 18], we report F1 and false positive rate (FPR) and assume that the set of languages covered by the evaluation benchmark is known. Accordingly, we restrict a model's predictions to those languages that occur in the intersection of the benchmark and model training data.

### 2.2.1 Evaluation data

We evaluate GlotLID v3.0 on three existing benchmarks with a high number of languages: UDHR, FLORES-200 and GlotTest (our in-domain test set).

1) **GlotTest:** Let $n_l$ be the number of sentences from language $l$ in the GlotLID corpus test set. Then we sample $\min(1000, n_l)$ sentences from it. We refer to the resulting dataset as GlotTest. GlotTest supports 2102 LID labels, the same number of labels supported by GlotLID v3.0. This includes the "und" labels (§2.1.2) and "zxx" labels (§2.1.3).

2) **UDHR:** UDHR consists of about 500 translations of the "Universal Declaration of Human Rights". In this work, we use the UDHR test set released in our previous work [37].[5] It supports the 415 translations from udhrinunicode.org that are available with a valid ISO 639-3 code. 371 of these 415 are covered by GlotLID v3.0 training data.

3) **FLORES-200:** FLORES-200 [55] comprises 842 articles sourced from English-language Wikimedia projects. Each sentence of these articles was translated into 204 language_script labels. The dataset is split into 997 sentences for development, 1012 for dev-test and 992 for test. FLORES-200 test is not public. As is common practice, we use FLORES-200 dev-test as our FLORES-200 test set.

---

[4]github.com/cisnlp/GlotLID/blob/main/sources.md
[5]hf.co/datasets/cis-lmu/udhr-lid

### 2.2.2 Evaluation results

Table 2 reports results on GlotTest, UDHR and FLORES-200. On average, GlotLID v3.0 achieves an F1 score of 0.991 and a false positive rate of 0.000003 on GlotTest. The three lowest F1 scores on GlotTest are 0.72, 0.75, and 0.76 for bos_Latn, hrv_Latn, and cnr_Latn, three mutually intelligible varieties of BCMS (Bosnian-Croatian-Montenegrin-Serbian). We made sure that every LID label has performance of at least 0.7 on the GlotLID validation set. Otherwise, we remove or merge the label (see §2.1.4).

Compared to GlotLID v1.0 [37], GlotLID v3.0 shows improvements in F1 of 0.05 for GlotTest, 0.09 for UDHR, and 0.05 for FLORES-200. Although the UDHR results are the lowest among these three benchmarks (F1 of 0.882 vs 0.991 and 0.967), GlotLID v3.0 outperforms the state-of-the-art also for this dataset [37]. The most likely reason for the lower performance on UDHR is a domain shift: the data in the GlotLID corpus for some of the UDHR languages is dominated by religious texts.

Table 2: Performance of GlotLID v3.0

| Benchmark | # Labels | F1 ↑ | FPR ↓ |
|---|---|---|---|
| GlotTest | 2102 | 0.991 | 0.000003 |
| UDHR | 371 | 0.882 | 0.000298 |
| FLORES-200 | 199 | 0.967 | 0.000161 |

## 3 GlotCC v1.0

The process of GlotCC creation is similar to other pipelines for large-scale web corpus creation [42]. More specifically, we use Ungoliant [3], the OSCAR pipeline [58]. Instead of OSCAR's FastText LID (which detects 176 languages), we use GlotLID v3.0. Recall that this change also includes the zxx and und labels that are part of GlotLID's label set and support robust noise removal (§2). We further provide extensions for content classes (§3.1) and quality warnings and filtering (§3.2, §3.3). We also perform replacement of personally identifiable information (§3.5) for GlotCC. We distribute our forked pipeline under the Apache 2.0 license, the same as the Ungoliant license. GlotCC is licensed under CommonCrawl terms: `commoncrawl.org/terms-of-use`.

For this paper, we ran the pipeline on the CommonCrawl CC-MAIN-2024-10 snapshot in its entirety and on parts of CC-MAIN-2023-40 and CC-MAIN-2023-50. We refer to the corpus produced by this process as GlotCC v1.0 (or GlotCC for short).

GlotCC v1.0 contains data (subcorpora) for 1275 LID labels (i.e., language-script pairs such as "rus-Cyrl"). Table 3 shows their geographic distribution for Glottolog [29] macroareas.[6] Based on the Wikipedia list of ISO 639-3 codes,[7] we also add 12 constructed languages ("Constructed"). As reflected in Table 4, GlotCC v1.0 considerably increases language coverage compared to OSCAR 23.01, especially for minority languages. The number of languages with more than $10^2$ documents in GlotCC is 145+89+52+29+22+12=349 (vs 132 for OSCAR), and with more than 10 documents is 349+360=709 (vs 142 for OSCAR). This language coverage can be easily increased by applying GlotLID on more CommonCrawl snapshots. One reason for GlotCC's better coverage could be that we lose less minority language content via contamination [13] to "large" languages. GlotCC's Wikipedia percentage is highest (.2658, .2940) for languages with a document count between $10^2$ and $10^4$. Many GlotCC languages with $\leq 10^2$ documents come from religious websites (.4441, .4285). GlotCC's coverage of languages with more than $10^7$ documents is lower than OSCAR's because we apply more cleaning filters.

Table 3: Geographic distribution of languages in GlotCC.

| Macroarea | # Labels |
|---|---|
| Eurasia | 395 |
| Papunesia | 380 |
| Africa | 252 |
| North America | 123 |
| South America | 97 |
| Australia | 16 |
| Constructed | 12 |

---

[6]"Papunesia" refers to "Insular" Southeast Asia and Oceania, excluding Australia.
[7]`wikipedia.org/wiki/Codes_for_constructed_languages`

Table 4: Partition statistics for OSCAR 23.01 and GlotCC-v1.0. Each partition is defined as: $10^J >$ # documents per language $\geq 10^I$ where $0 \leq I \leq 7$, $1 \leq J \leq 9$.

| {$I, J$} | Corpus Version | # Languages | # Documents | | # Lines | | # Words | | # Religious | # Wikipedia |
|---|---|---|---|---|---|---|---|---|---|---|
| | | | Total | Median | Total | Median | Total | Median | Total pct. | Total pct. |
| {7, 9} | OSCAR 23.01 | 24 | 2.7B | 34.4M | - | - | 1.0T | 12.6B | - | - |
| | GlotCC-v1.0 | 12 | 579.5M | 22.7M | 15.1B | 780.8M | 436.4B | 17.0B | 0.0001 | 0.0009 |
| {6, 7} | OSCAR 23.01 | 23 | 80.0M | 2.4M | - | - | 27.6B | 738.8M | - | - |
| | GlotCC-v1.0 | 22 | 92.2M | 3.8M | 3.0B | 122.1M | 67.8B | 2.4B | 0.0001 | 0.0044 |
| {5, 6} | OSCAR 23.01 | 25 | 9.3M | 262.7K | - | - | 3.2B | 82.4M | - | - |
| | GlotCC-v1.0 | 29 | 10.7M | 334.8K | 305.4M | 9.1M | 6.9B | 195.7M | 0.0001 | 0.0219 |
| {4, 5} | OSCAR 23.01 | 26 | 919.7K | 25.2K | - | - | 212.0M | 5.4M | - | - |
| | GlotCC-v1.0 | 52 | 1.9M | 29.6K | 55.1M | 714.4K | 1.3B | 17.9M | 0.0005 | 0.0922 |
| {3, 4} | OSCAR 23.01 | 14 | 60.1K | 3.6K | - | - | 10.1M | 315.7K | - | - |
| | GlotCC-v1.0 | 89 | 338.7K | 2.7K | 8.2M | 52.2K | 223.9M | 1.4M | 0.0029 | 0.2658 |
| {2, 3} | OSCAR 23.01 | 20 | 8.6K | 400 | - | - | 772.3K | 13.4K | - | - |
| | GlotCC-v1.0 | 145 | 53.9K | 326 | 1.4M | 6.5K | 39.3M | 192.6K | 0.0606 | 0.2940 |
| {1, 2} | OSCAR 23.01 | 10 | 368 | 36 | - | - | 13.6K | 431 | - | - |
| | GlotCC-v1.0 | 360 | 11.5K | 24 | 245.0K | 460 | 11.3M | 20.5K | 0.4441 | 0.1044 |
| {0, 1} | OSCAR 23.01 | 10 | 44 | 4 | - | - | 21.5K | 67 | - | - |
| | GlotCC-v1.0 | 566 | 1.7K | 2 | 41.5K | 26 | 1.7M | 1.2K | 0.4285 | 0.0285 |
| {0, 9} | OSCAR 23.01 | 152 | 2.8B | 69.7K | - | - | 1.1T | 14.5M | - | - |
| | GlotCC-v1.0 | 1275 | 684.7M | 14 | 18.5B | 254 | 512.6B | 11.6K | 0.000001 | 0.00000007 |

## 3.1 Annotating documents with content classes

Similar to OSCAR 23.01, we use the UT1 blocklist [61] to classify websites into different content classes such as "adult" and "blog". The main utility of these UT1-based filters is to warn about potential adult content websites, with 3.5 million domains labeled as adult in UT1.

We add to UT1 two additional content classes: "wikipedia" and "religious".[8] The religious content class is important to assess how domain-specific a particular language's corpus is – for some minority languages almost all web content is religious [42]. Following OSCAR 23.01, we do not remove any content classes from GlotCC and instead leave that decision (e.g., removal of adult content) to the user, mainly because of UT1 false positives [4].

## 3.2 Quality warnings

We add new quality warnings adopted from prior work on data cleaning and web crawling [63, 62, 4, 30, 40, 42]. Specifically, we provide the following quality warnings.[9]

**Tiny**: The document has a small number of lines. Following [63], we use a threshold of three lines.

**Short sentences**: The document has a high number ($\geq 50\%$) of short lines [4].

**Header and footer**: CommonCrawl contains boilerplate extracted from headers and footers. We give a *header* (resp. *footer*) warning if short lines occur at the start (resp. end) of the document [4].

**Inconsistent**: Ungoliant applies LID at both document and line levels. If $\geq 60\%$ of lines do not match the document-level label, we mark the document as *LID-inconsistent*. If $\geq 10\%$ of the script content is incompatible with the label predicted by LID, we mark the document as *script-inconsistent* [40].

**List case**: Flag sentences with $\geq 50\%$ of tokens beginning with a capital letter [42]. Some scripts like Hani lack capital letters. However, since Chinese is written without spaces, we can still find lists by identifying portions of the text that are shorter than five characters and are surrounded by spaces.

**Technical characters**: Fires when $\geq 20\%$ of characters are numbers/punctuation [42]. The warning *script-inconsistent* is also raised here since $\geq 10\%$ is not written in the main script.

**Cursed regex**: These are substrings and regexes from [42] used for identifying noisy and questionable content.

---

[8]We compile the religious category based on popular websites like `jw.org` and `ebible.com`. While we believe it to be a helpful indicator for the user, it is not perfect, neither in terms of recall (especially for non-Christian content) nor precision (many religious websites also contain non-religious content).

[9]In the following discussion, we use the terms "line" and "sentence" interchangeably.

**Repetition**: Repetition of words and bigrams indicates poor quality [62]. We give the warning *repetition* if a sentence has >20 words and either >50% of the words or >20% of the bigrams are repetitive. This heuristic cannot be applied to languages without word boundaries.

**Long word**: This warning applies if there is a word with more than 100 characters; see [42].

**Lorem ipsum**: If the text contains the placeholder "lorem ipsum" [63], we flag it as *lorem ipsum*.

**Policy**: Many texts have boilerplate policy notices [63]. We flag the text as *policy* if it contains any of the following: "terms of use", "privacy policy", "cookie policy", "uses cookies", "use of cookies", "use cookies".

**JS warning**: Many texts contain warnings that JavaScript should be enabled [63]. If the text contains "JavaScript" or "Javascript", we flag it as *js warning*.

**Curly bracket**: Curly braces are mostly used in programming languages, not natural language [63]. If the text contains curly braces, we flag it as *curly bracket*.

**Adult words**: This is a collection of pornographic keywords from [42], mostly for Chinese tokens. We also found out that the GPT-4o tokenizer [57] has many adult tokens and gambling terms, so we conducted a manual audit of Chinese tokens longer than 3 characters in this tokenizer's vocabulary and added them as additional keywords.

### 3.3 Quality warning filters

We sample 20 sentences from three languages for each script: one from the top 10% of the language distribution (i.e., with most data in GlotCC), one from the bottom 75% and one from the remaining 15%. For each sentence, we determine whether it is high-quality content in the target language. For unknown languages, we check the website URL and search for information about the language. We find that the quality warnings generally indicate bad content or erroneously assigned LID labels and therefore remove sentences with quality warnings in GlotCC. There are two exceptions.

First, we ignore three warnings: *short sentences*, *header*, and *footer*, because the LID label is correct for most of these documents, or in other cases, the warnings are false positives. Second, for languages without overt word boundaries,[10] we keep documents with warnings *long word* and *repetition* as computing these warnings is nonsensical if the text is not separated into words. The warning metadata for these five warnings is kept in GlotCC in case users want to use it.

### 3.4 Deduplication

Following OSCAR 23.01, a hash is provided for each document in GlotCC. This hash is computed by `py-tlsh`[11] with hyperparameters 256 buckets and 3-byte checksums [4]. However, we do not distribute deduplication as part of the pipeline, because deduplication is costly [26, 4] and hashing algorithm and hyperparameters are application-dependent.

### 3.5 Personally identifiable information replacement

We replace two types of personally identifiable information (PII): email addresses and public network IP addresses. We do not replace phone numbers due to the high false positive rate of regex patterns. We use the PII process implemented by DataTrove [59] (see also FineWeb [60]). Email addresses are replaced with "email@example.com" or "firstname.lastname@example.com," and public network IP addresses are replaced with one of six IP addresses: "22.214.171.124," "126.96.36.199," "188.8.131.52," "184.108.40.206," "220.127.116.11," or "18.104.22.168," which, at the time of corpus creation, were unresponsive to pings.

### 3.6 Wall time

We calculate the pipeline's wall time, considering only the LID, without accounting for quality warning filters, the PII process, or any infrastructure-related bottlenecks. Suppose the LID throughput is, on average, $T_S$ sentences per second and $T_D$ documents per second, and we can run $P$ parallel jobs.

---

[10]`wikipedia.org/wiki/Category:Writing_systems_without_word_boundaries`
[11]`pypi.org/project/py-tlsh`

We have $D$ documents to annotate, each containing an average of $S$ sentences. For each document, in addition to running the LID on the entire document, we also run it on each individual sentence. The estimated processing time in hours is given by:

$$\text{Wall time (hours)} = \frac{D}{3600 \times P} \times \left( \frac{S}{T_S} + \frac{1}{T_D} \right)$$

The processing time on Intel Xeon E7-8857 3GHz CPUs with $P = 48$ for one Common Crawl snapshot ($D = 3.16 \times 10^9$ for CC-MAIN-2024-10) is estimated, using the given values of $T_S = 1379$, $T_D = 245$, and $S = 20$, to be approximately 340 hours.

### 3.7 Self-audit quality review

We perform a self-audit of GlotCC-V1.0. We have two motivations. First, following [41], we want to ensure that the target language metadata are correct and that there are no systematic issues. Second, we intend to develop additional filters to clean the corpus for future releases. The bottleneck in such an audit is the difficulty of finding native speakers for each language. Therefore, following [42], we conduct the audit by providing high-level comments on the data quality and identifying the language of the data by looking for language clues.

We randomly select 653 languages. We sample 20 pages from each (or all if there are fewer than 20) and check the validitiy of GlotCC's LID label. Additionally, we analyze common errors and provide high-level comments.

We follow these guidelines in conducting the audit:

- For unknown languages, we inspect the URL and visit the webpage to find language clues such as language codes (especially in the URL), the `lang` attribute inside the html tag, country flag, contact address and the name of the language in the text. Otherwise, we search for sentences on the web to consult similar webpages related to that sentence.
- If the corpus contains noise but the noise appears filterable, we leave a high-level note detailing the noise and how it can be filtered.
- We report the percentage of in-language sentences for each audited language.

**Overall results.** Out of 653 audited languages, we find that, with a macro-average score of 0.93 and a median score of 1.0, the data is in-language. During the audit, for some languages, we couldn't determine the correct language; therefore, we do not consider those languages in the audited set. We find 10 out-of-model languages: aon (Bumbita Arapesh, Torricelli), bkx (Baikeno, Malayo-Polynesian), gup (Gunwinggu, Arnhem, Northern Australia), ibl (Ibaloi, Philippine), kpo (Ikposo, Atlantic-Congo), mcr (Menya, Trans-New Guinea), mge (Mango, Nilo-Saharan), mrh (Mara Chin, Sino-Tibetan), sxw (Saxwe Gbe, Atlantic-Congo) and tao (Yami, Malayo Polynesian). We plan to include these languages in GlotLID v4.0. There are also errors that neither the LID nor the filters captures. For example, repetitive n-grams in list-like content, such as those at the start or end of words from websites like `anagrams.app`. Based on these audits, we published a more clean version of GlotCC-V1.0 to the community.

### 3.8 Evaluation of LID within the pipeline

We compare the NLLB LID [55] and GlotLID within the context of the Ungoliant pipeline. For this comparison, we run the pipeline in the exact same configuration, except that we use NLLB LID for one run and GlotLID for the other. We look at a random sample of minority language pages for which the two pipelines make different predictions.

In more detail, we select a subset of size $\approx$1% (a prefix) of the latest CommonCrawl snapshot (CC-MAIN-2024-18) and run the GlotCC pipeline on it, once using GlotLID as the LID and once using the NLLB LID as the LID. This produces a corpus that is analogous to GlotCC-v1.0, except it is based on (a prefix of) a different CommonCrawl snapshot. We count the number of times that an LID label (e.g., rus-Cyrl) is assigned by either GlotLID or NLLB LID in this "filter" subset; we refer to this number as $n_l$ for label $l$. We restrict the comparison to those labels that occur $n_l \leq 10$ times because our main focus is minority languages, not "large" languages that are already well covered by existing resources. We further only consider pages for which GlotLID and NLLB LID disagree. This

gives us a final set of 260 pages. Note that this set of 260 pages would correspond to $100 \times 100 \times 260$ = 2,600,000 pages, had we run the pipeline on the entire CommonCrawl. (We take a prefix of size 1% of the latest snapshot and there have been about 100 CommonCrawl snapshots so far.)

Table 5 reports a comparison of GlotLID and NLLB LID for a random sample of 20 of the 260 pages; see codebase for a detailed report of the comparison. "miss" refers to cases where the LID did not make a call. We see that GlotLID is correct 13 times for pages that NLLB LID treated incorrectly (3 misclassified pages, 10 "miss" pages) and incorrect 5 times for correct decisions by NLLB LID (3 incorrect calls and 2 incorrect "miss" decisions). On 2 pages, both GlotLID and NLLB LID make incorrect decisions.

Table 5: Comparison of GlotLID and NLLB on a random subset of 20 pages from minority languages

| | NLLB→ | correct | error | miss |
|---|---|---|---|---|
| GlotLID ↓ | | | | |
| correct | | 0 | 3 | 10 |
| error | | 3 | 2 | 0 |
| miss | | 2 | 0 | - |

The main reason for GlotLID's clearly better performance is that it makes many more calls than the prior state of the art, without losing overall accuracy. This is in keeping with the substantial expansion of GlotLID's label set (more than 2,000) compared to prior work.

## 4   Related work

### 4.1   LID

There is a wealth of resources to perform LID, but not all of them meet the requirements of a good LID for minority corpus creation [37]. All of the following cover at most 218 languages: CLD2 [50], Equilid [36], Langdetect [65], langid.py [47], OpenLID [18], NLLB LID [55], FastText LID [14], CLD3 [15, 64], and HeLI-OTS [33]. Some LIDs are not open-source, e.g., those published by Caswell et al. [20], Bapna et al. [11], Kudugunta et al. [42]. whatlang [16, 17] and idNet [23] are two broad-coverage LIDs that meet many other requirements but are hard to use in many practical scenarios due to software issues and lack of maintenance. Franc [72] is a character 3-gram LID with support for 400+ languages; however, it does not provide well-calibrated probabilities. Another LID with similar properties and support for 1600+ languages is FUN-LangID [71]; however, according to the developers, this model is not the best for high performance on F1/FPR. AfroLID [5], which covers African languages, is a Transformer architecture and less efficient than its competitors. A strong requirement for LIDs is their effective applicability for very large corpora, not least for ecological reasons. GeoLID [24] meets most of the requirements; it supports 900+ languages and the architecture is based on FastText. However, it needs geographic prior information, which makes it more suitable for social media such as X (Twitter) that provide such geographic priors.

### 4.2   Multilingual corpora

Much work has been done on mining multilingual corpora from the web. Xue et al. [73] introduce mC4, a general 101-language web domain corpus, to train the mT5 model. Similarly, Conneau et al. [22] introduce CC-100 based on the CC-Net repository [69] to train the XLM-R model. The OSCAR corpus [4] supports 150+ languages. The mC4 pipeline uses CLD3, and CC-Net and OSCAR use FastText LID. NLLB Team et al. [55] mine an internal corpus from CommonCrawl with 200+ languages using NLLB LID. The MADLAD-400 corpus [42] is another mined corpus with 450+ languages using an internal 500-language coverage LID and pipeline. The most closely related work to ours is by Bapna et al. [11], who create an internal corpus of 1500+ languages using an internal 1600+ LID and pipeline. There are also high-quality pipelines that focus on creating corpora mostly for English, including Dolma [67], RedPajama-Data-v2 [21], DCLM [46] and FineWeb [60]. There is also some work that does not only mine the web, but builds mulitlingual data upon other datasets by compiling multiple data sources, such as the ROOTS corpus [44], a community-built dataset that contains 46 languages. CulturaX [53] combines mC4 3.1.0 and different OSCAR versions and applies additional filters, resulting in 160+ languages. The Glot500 corpus [30] covers 500+ languages (400+ open-access), mostly based on prior work in academia. Serengeti [6] also introduces an internal dataset of 500+ African languages derived from religious texts, news and academia.

To the best of our knowledge, GlotCC is the first open-access corpus that is based on an open pipeline (including open-access LID) and that passes the threshold of 160 languages (e.g., CulturaX [53]) in web-mined corpora; in fact, as we show we cover more than a thousand languages. As our statistics show, we find many languages on the web that have never before been part of a web-mined corpus.

## 5 Limitations

**Use cases:** Due to certain filtering steps (e.g., curly bracket filter), the GlotCC likely does not contain much math/code content. It is advisable to supplement GlotCC with math/code data if this is the intended use case. Due to the use of LID and script inconsistency filters as indicators of noise, the documents that exist in the corpus are more monolingual than multilingual or code-switched [39] compared to when these filters are not used. Additionally, since we did not customize the processing for each website, some sources, such as Wikipedia, may have better formatting in the original than in GlotCC.

**Errors:** Although we have good in-language content, GlotCC still exhibits many types of errors and noise, including misclassification and out-of-model cousins.

**Model training:** We did not train any language model to better justify the significance of GlotCC, as this would be pointless unless we also evaluate them. Evaluating language models requires evaluation data that we mostly do not have for minority languages (see [38, 9, 8, 30]). At this stage, this endeavor seems unrealistic. We believe that identifying relevant data and performing LID is a crucial first step in that direction.

## 6 Conclusion

We introduce a document-level, general domain web corpus covering more than 1000 languages. We open-source the entire pipeline, including a better language identification model, which is more robust in the corpus creation task in terms of noise handling, unseen writing systems, and a broad coverage of languages to reduce the chance of encountering unknown languages. We filter the created corpus and perform a self-audit to ensure the created corpus is clean.

We hope the creation of such a corpus and pipeline will benefit language technologies, enabling the inclusion of more minority languages. For future work, we plan to extend this corpus to additional CommonCrawl snapshots.

## 7 Ethics statement

The advancement of NLP technologies has primarily been constrained to languages for which resources are available in good quality and quantity. Many of the world's minority languages face a significant barrier due to the scarcity of high-quality general data sources, making it difficult to develop NLP tools for these languages. Despite concerns that an "extractive" approach to NLP often does not benefit the affected communities [12], it is still an important goal of both computational and theoretical linguistics to have as good a representation of minority languages in available web-based corpora as is permitted by the licenses of content available on the web. By providing a cleaned corpus like GlotCC, we take initial steps towards including a diverse range of languages in NLP. Despite our strong focus on filtering, the cleaning of GlotCC is constrained by the lack of tools for filtering out undesirable content and noise – such as adult content, personal information, lists – and this is a considerable problem for a subset of languages, including languages without explicit word boundaries. Therefore, we recommend users carefully evaluate their specific use case before using GlotCC.

### Acknowledgments and disclosure of funding

We would like to thank anonymous reviewers. This work was funded by Deutsche Forschungsgemeinschaft (project SCHU 2246/14-1).

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

# A  Appendix

## A.1  GlotLID model card

We provide the GlotLID v3.0 model card based on the model card template introduced by Mitchell et al. [52].

---

**Model details**

- Person or organization developing model: *LMU Munich and Sorbonne Université*

- Model Date: *April 18, 2024*

- Model Types: *Language identification and language modeling.*

- Model Access: ○ github.com/cisnlp/GlotLID and 🤗 hf.co/cis-lmu/glotlid

- Information about training algorithms, parameters, fairness constraints or other applied approaches, and features: *Provided in the Section 2*

- Paper:

    - V3: *Kargaran et al, GlotCC: An Open Broad-Coverage CommonCrawl Corpus and Pipeline for Minority Languages, NeurIPS, 2024*
    - V1: *Kargaran et al, GlotLID: Language Identification for Low-Resource Languages, EMNLP, 2023*

- License: *Apache License 2.0*

- Contact: *amir@cis.lmu.de*

**Intended use**

- Primary intended uses: *Language identification on over 2000 linguistic labels.*

- Primary intended users: *Research community.*

- Out-of-scope use cases: *The model is trained on general domain data and might not perform well with short sentences or domain-specific data like the medical domain.*

**Factors**

- *The classification quality of this model varies based on language, as some languages are easy to distinguish while others are challenging with n-gram models.*

**Metrics**

- *We use F1 score and FPR (false positive rate), two widely used metrics in the language identification literature, for our evaluations.*

**Ethical considerations**

- *We acknowledge that this model has a high error rate for some of the languages. This means that there is a potential risk of excluding minority languages during the collection and processing of NLP corpora.*

**Training data**

- *We train this model using openly available (but not necessarily freely redistributable) datasets, including resources previously published by researchers, publishers, and translators. The complete list of data sources is available on* ○ *github.com/cisnlp/GlotLID/blob/main/sources.md.*

**Evaluation data**

- *For evaluation, we used GlotTest, UDHR, and Flores-200 as described in §2.2.1.*

**Caveats and recommendations**

- *We note that if there is a setup where the list of languages is known, then the model can limit its predictions to that set of languages. This implies that languages not included in the set will be excluded from the softmax computation. We have provided the code for limiting this on* ○ *github.com/cisnlp/GlotLID.*

---

## A.2 GlotCC datasheet

We provide the GlotCC v1.0 datasheet based on the datasheet template introduced by Gebru et al. [27].

### Motivation

1. For what purpose was the dataset created? (Was there a specific task in mind? Was there a specific gap that needed to be filled? Please provide a description.) *We created GlotCC as a general web-crawled document-level dataset covering 1275 languages, with the purpose of breaking barriers by providing training data for language technologies to include a broader range of languages.*

2. Who created this dataset and on behalf of which entity? *Amir Hossein Kargaran♣, François Yvon♠, Hinrich Schütze♣ (♣LMU Munich, ♠Sorbonne Université)*

3. Who funded the creation of the dataset? *DFG (grant SCHU 2246/14-1).*

4. Any other comments? *None.*

### Composition

1. What do the instances that comprise the dataset represent? *Each instance is a filtered instance from CommonCrawl, with its language annotated by GlotLID.*

2. How many instances are there in total? *GlotCC has 684.7M documents (18.5B lines, or 512.6B words) total across 1275 languages.*

3. Does the dataset contain all possible instances or is it a sample (not necessarily random) of instances from a larger set? (If the dataset is a sample, then what is the larger set? Is the sample representative of the larger set (e.g., geographic coverage)? If so, please describe how this representativeness was validated/verified. If it is not representative of the larger set, please describe why not (e.g., to cover a more diverse range of instances, because instances were withheld or unavailable).) *GlotCC is created from CommonCrawl and has been annotated by GlotLID, then filtered. To maintain a high level of in-language content, we have employed aggressive filtering, which may result in the exclusion of certain documents in a specific language within CommonCrawl.*

4. What data does each instance consist of? *Each instance is a raw text with accompanying metadata, including the timestamp, URL, quality warnings, category, tlsh hash, language identification probability, language identification consistency score, script consistency score, number of sentences, and content length.*

5. Is there a label or target associated with each instance? If so, please provide a description. *Yes, GlotCC has a language label for each instance.*

6. Is any information missing from individual instances? *No.*

7. Are relationships between individual instances made explicit? *No.*

8. Are there recommended data splits (e.g., training, development/validation, testing)? *No.*

9. Are there any errors, sources of noise, or redundancies in the dataset? *Although GlotCC have good in-language content, GlotCC still exhibits many types of errors and noise, including misclassification and out-of-model cousins.*

10. Is the dataset self-contained, or does it link to or otherwise rely on external resources (e.g., websites, tweets, other datasets)? (If it links to or relies on external resources, a) are there guarantees that they will exist, and remain constant, over time; b) are there official archival versions of the complete dataset (i.e., including the external resources as they existed at the time the dataset was created); c) are there any restrictions (e.g., licenses, fees) associated with any of the external resources that might apply to a future user? Please provide descriptions of all external resources and any restrictions associated with them, as well as links or other access points, as appropriate.) *Yes.*

11. Does the dataset contain data that might be considered confidential (e.g., data that is protected by legal privilege or by doctor-patient confidentiality, data that includes the content of individuals' non-public communications)? (If so, please provide a description.) *It is possible as GlotCC is a general web-crawled dataset.*

12. Does the dataset contain data that, if viewed directly, might be offensive, insulting, threatening, or might otherwise cause anxiety? (If so, please describe why.) *It is possible as GlotCC is a general web-crawled dataset.*

13. Does the dataset relate to people? *It's possible that some instances of GlotCC mention and describe individuals.*

14. Does the dataset identify any subpopulations (e.g., by age, gender)? *It's possible that some instances of GlotCC mention and describe people of certain subpopulations.*

15. Is it possible to identify individuals (i.e., one or more natural persons), either directly or indirectly (i.e., in combination with other data) from the dataset? *It's possible that some instances of GlotCC mention and describe individuals.*

16. Does the dataset contain data that might be considered sensitive in any way (e.g., data that reveals racial or ethnic origins, sexual orientations, religious beliefs, political opinions or union memberships, or locations; financial or health data; biometric or genetic data; forms of government identification, such as social security numbers; criminal history)? *It is possible as GlotCC is a general web-crawled dataset.*

17. Any other comments? *None.*

## Collection

1. How was the data associated with each instance acquired? *From CommonCrawl CC-MAIN-2024-10 snapshot eniterly and on parts from CommonCrawl CC-MAIN-2023-40 and CC-MAIN-2023-50.*

2. What mechanisms or procedures were used to collect the data (e.g., hardware apparatus or sensor, manual human curation, software program, software API)? *We annotated the CommonCrawl data using GlotLID and then filtered the documents to create GlotCC.*

3. If the dataset is a sample from a larger set, what was the sampling strategy? *GlotCC is a subset of CommonCrawl documents based on the GlotLID annotations and filtering steps.*

4. Who was involved in the data collection process (e.g., students, crowdworkers, contractors) and how were they compensated (e.g., how much were crowdworkers paid)? *For the audit, the authors inspected the dataset.*

5. Over what timeframe was the data collected? (Does this timeframe match the creation timeframe of the data associated with the instances (e.g., recent crawl of old news articles)? If not, please describe the timeframe in which the data associated with the instances was created.) *Each instance is followed by a UTC timestamp that shows the time of the crawl provided by CommonCrawl.*

6. Were any ethical review processes conducted (e.g., by an institutional review board)? *No.*

7. Does the dataset relate to people? *It's possible that some instances of GlotCC mention and describe individuals.*

8. Did you collect the data from the individuals in question directly, or obtain it via third parties or other sources (e.g., websites)? *We collect the data from CommonCrawl.*

9. Were the individuals in question notified about the data collection? *No.*

10. Did the individuals in question consent to the collection and use of their data? *No.*

11. If consent was obtained, were the consenting individuals provided with a mechanism to revoke their consent in the future or for certain uses? *N/A.*

12. Has an analysis of the potential impact of the dataset and its use on data subjects (e.g., a data protection impact analysis) been conducted? *No.*

13. Any other comments? *None.*

## Preprocessing/cleaning/labeling

1. Was any preprocessing/cleaning/labeling of the data done (e.g., discretization or bucketing, tokenization, part-of-speech tagging, SIFT feature extraction, removal of instances, processing of missing values)? (If so, please provide a description. If not, you may skip the remainder of the questions in this section.) *We filter the data based on various quality warning filters.*

2. Was the "raw" data saved in addition to the preprocessed/cleaned/labeled data (e.g., to support unanticipated future uses)? *We do not change the raw data; we only provide the metadata and determine which CommonCrawl instances to keep.*

3. Is the software used to preprocess/clean/label the instances available? *Yes,* ⚬ `github. com/ cisnlp/ GlotCC`

4. Any other comments? *None.*

## Uses

1. Has the dataset been used for any tasks already? (If so, please provide a description.) *No.*

2. Is there a repository that links to any or all papers or systems that use the dataset? *No.*

3. What (other) tasks could the dataset be used for? *GlotCC can serve as a general training dataset for any language included in GlotCC.*

4. Is there anything about the composition of the dataset or the way it was collected and preprocessed/cleaned/labeled that might impact future uses? (For example, is there anything that a future user might need to know to avoid uses that could result in unfair treatment of individuals or groups (e.g., stereotyping, quality of service issues) or other undesirable harms (e.g., financial harms, legal risks) If so, please provide a description. Is there anything a future user could do to mitigate these undesirable harms?) *Despite our strong focus on filtering, the cleaning of GlotCC is constrained by the lack of tools for filtering out undesirable content and noise – such as adult content, personal information, lists – and this is a considerable problem for a subset of languages, including languages without explicit word boundaries. Therefore, we recommend users carefully evaluate their specific use case before using GlotCC.*

5. Are there tasks for which the dataset should not be used? (If so, please provide a description.) *N/A.*

6. Any other comments? *None.*

**Distribution**

1. How will the dataset will be distributed (e.g., tarball on website, API, GitHub)? *GlotCC is made available through a Huggingface datasets* 🤗 `hf.co/datasets/cis-lmu/GlotCC-V1`.

2. When will the dataset be distributed? *June 2024.*

3. Will the dataset be distributed under a copyright or other intellectual property (IP) license, and/or under applicable terms of use (ToU)? (If so, please describe this license and/or ToU, and provide a link or other access point to, or otherwise reproduce, any relevant licensing terms or ToU, as well as any fees associated with these restrictions.) *LMU Munich hosts GlotCC on Huggingface. We license the GlotCC metadata under CC0 1.0 (public domain). However, the GlotCC is licensed under the terms of the CommonCrawl Terms of Use.*

4. Have any third parties imposed IP-based or other restrictions on the data associated with the instances? *GlotCC is licensed under the terms of the CommonCrawl Terms of Use.*

5. Any other comments? *None.*

**Maintenance**

1. Who is supporting/hosting/maintaining the dataset? *LMU Munich on Huggingface.*

2. How can the owner/curator/manager of the dataset be contacted (e.g., email address)? *Amir Hossein Kargaran* (`amir@cis.lmu.de`).

3. Is there an erratum? (If so, please provide a link or other access point.) ⬡ `github.com/cisnlp/GlotCC`

4. Will the dataset be updated (e.g., to correct labeling errors, add new instances, delete instances')? (If so, please describe how often, by whom, and how updates will be communicated to users (e.g., mailing list, GitHub)?) *Yes, we maintain the data based on discussions on Huggingface, GitHub and those reported via email.*

5. If the dataset relates to people, are there applicable limits on the retention of the data associated with the instances (e.g., were individuals in question told that their data would be retained for a fixed period of time and then deleted)? (If so, please describe these limits and explain how they will be enforced.) *N/A.*

6. Will older versions of the dataset continue to be supported/hosted/maintained? (If so, please describe how. If not, please describe how its obsolescence will be communicated to users.) *Yes, any version of the data will be hosted for the sake of reproducibility of results for others using the dataset.*

7. If others want to extend/augment/build on/contribute to the dataset, is there a mechanism for them to do so? (If so, please provide a description. Will these contributions be validated/verified? If so, please describe how. If not, why not? Is there a process for communicating/distributing these contributions to other users? If so, please provide a description.) *We made the system to generate GlotCC, including the pipeline, language identification model, and filters, available to the research community. This allows them to build upon it or customize it for their use case.*

8. Any other comments? *None.*

