# OpenReview forum: "GlotCC: An Open Broad-Coverage CommonCrawl Corpus and Pipeline for Minority Languages"
_NeurIPS.cc/2024/Datasets_and_Benchmarks_Track — NeurIPS 2024 Track Datasets and Benchmarks Poster_

### Official Review · Reviewer_6BqC · 2024-07-25
**Novel multilingual pipeline, limited evaluation of downstream corpus**

**Rating:** 6
**Confidence:** 4
**Correctness:** The work is rigorous and I could not …
**Clarity:** The manuscript is well written and ea…

**Review:**

Overall, the language identification approach presented in this work is well thought out and effective on intrinsic benchmarks. The proposed method employs both principled approaches to tackle low-resource languages, as well as well engineered tricks to offset classes of frequent mistakes.

On the other hand, its application to curating GlotCC is not well evaluated. The manuscript lacks any evaluation for this corpus, either through intrinsic metrics, or downstream applications, such as language modeling.

**Strengths:**

- This work puts great care in increasing coverage of languages that can be identified on web text.
- The proposed approach increases coverage of languages that can be identified without decreasing computational efficiency.
- The manuscript present GlotCC, a large scale web dataset containing documents in over a thousand languages.

**Additional Feedback:**

N/A

**Documentation:**

Documentation is provided on Huggingface; Pipeline is documented on github.

**Ethics:**

No ethics violations to report.

**Limitations:**

- Effectiveness of steps used to curate GlotCC are not validated in any way. We don't know how the resulting corpus compares to MADLAD, CulturaX, OSCAR, … or similar efforts. Summary statistics of the major related corpora would be a great start; further, an estimation of relative quality (either through some intrinsic measure, or via language modeling) would be useful.
- The accuracy of GlotLID is not measured on documents selected in GlotCC, except in Table 3. Since no statistically significant conclusion can be drawn from just 20 random samples, I would have liked to see a more comprehensive evaluation.
- Section 3.4: The proposed pipeline does not perform deduplication due to its computational cost. However, the pipeline could have taken advantage of more efficient deduplication pipeline, such as a bloom-filter based approach (BFF, Dolma), or use more efficient libraries than py-tlsh. Further, they could have partitioned the collection and deduplicated over multiple rounds on smaller subsets. Common crawl exhibits a high degree of duplication, so this step is fundamental when curating a dataset from it.

**Opportunities For Improvement:**

- Lines 25-27: *"For example, FastText’s accuracy suffers from hash collisions: it maps out-of-domain (OOD) n-grams to n-grams seen in training, resulting in many OOD errors"* FastText language identification is not accurate for many reasons, but authors should be more clear on why this specific failure mode is highlighted in the introduction.
- Section 2.1.2: When discussing the issue arising from undefined ngrams in fasttext, the authors could have evaluated a bloom filter implementation to reduce memory footprint for rare and unseen ngrams (e.g., Floret), or rely on featurization via BPE with byte fallback, as the `und_*` approach presented in this work requires knowing at priori which symbols are not present.
- Section 2.1.3: The manuscript could discuss how much of the 6 issues discussed here are specific to the HTML linearization pipeline and crawl used in this work. In other words, if other internet sources were to be used, would these rules be needed?
- Section 2.4: GlotLID v3 is not compared to any other language identification technique. For example, I would have expected a comparison with the MADLAD.
- Section 3: recent works have highlighted the importance of re-parsing HTML content using a modern parser (Trafilatura for RefinedWeb, Resiliparse for OpenWebMath). The paper could discuss potential issues that arise from relying on OSCAR pipeline instead, which uses WET files from Common Crawl.

**Relation To Prior Work:**

Prior work is acceptably discussed.

**Summary And Contributions:**

The authors introduce a new pipeline for extracting multilingual text from CommonCrawl. The pipeline is designed to address limitations of current LID models by (a) covering more languages, and (b) identify web noise so that it doesn't cause misidentification.

---

> ### Author Rebuttal · Authors · 2024-08-17
>
> Thank you for the detailed feedback on our work and for highlighting the increasing coverage of languages in our corpus without decreasing computational efficiency, as well as our focus on well-thought-out language identification. We have addressed the questions and concerns raised in your review below:
>
>
> **Re: Opportunities For Improvement**:
>
> > Lines_25-27:_"For_example,_FastText’s_accuracy_suffers_from_hash_collisions:_it_maps_out-of-domain_(OOD)_n-grams_to_n-grams_seen_in_training,_resulting_in_many_OOD_errors"....
>
>
> We introduce additional labels to GlotLID (GlotLID v3.0 vs. GlotLID v1.0/v2.0: more languages, und_ and zxx). The motivation behind these labels is to reduce OOD n-grams, thereby enabling a more reliable language identification (LID).
>
> > Section_2.1.2:_When_discussing_the_issue_arising_from_undefined_ngrams_in_fasttext,_the_authors_could_have_evaluated_a_bloom_filter_implementation_to_reduce_memory_footprint_for_rare_and_unseen_ngrams_(e.g.,_Floret),....
>
> When you support many more languages and scripts, as we do, the number of n-grams increases significantly, making it harder to manage.
> Floret is definitely an interesting project. However, as the fastText and floret binary formats are not compatible, we did not try Floret, as our pipeline is based on fastText [Rust binding](https://github.com/messense/fasttext-rs). We would like to incorporate this in the next version of our pipeline, however, this will require additional engineering to ensure compatibility.
>
> > Section_2.1.3:_The_manuscript_could_discuss_how_much_of_the_6_issues_discussed_here_are_specific_to_the_HTML_linearization_pipeline_and_crawl_used_in_this_work._...
>
> Thanks for pointing this out. These errors were mostly observed when working with CommonCrawl data, which is one of the most popular sources and is used for many great corpora. Other formats of internet data might be cleaner or might present different types of errors that we could still improve under the same scheme. If you have a specific resource in mind, we can conduct a short study.
>
>
> > Section_2.4:_GlotLID_v3_is_not_compared_to_any_other_language_identification_technique._For_example,_I_would_have_expected_a_comparison_with_the_MADLAD.
>
> MADLAD LID is not openly available for comparison.
>
> We have already evaluated GlotLID v1.0 against other baselines, such as Fasttext LID, CLD3, OpenLID, and NLLB LID, in our [previous work](https://arxiv.org/abs/2310.16248). Evaluating against other LIDs is not the focus of this paper.
>
> Are you suggesting applying GlotLID to the MADLAD-400 cleaned data? If so, we would arrange this for the camera-ready. We would expect to see some minor discrepancies that would require auditing to determine which is more accurate.
>
>
> > Section_3:_recent_works_have_highlighted_the_importance_of_re-parsing_HTML_content_using_a_modern_parser_(Trafilatura_for_RefinedWeb,....
>
> Thanks for pointing this out; we will add it to the paper. WET files take up less disk space and are easier to process. While Trafilatura is excellent, it is significantly more expensive if we want to run our experiments directly on WARC files. Moreover, we have to make sure that the new text extraction solution works for every language/script.
>
>
> **Re: Limitations**:
>
>
> > Effectiveness_of_steps_used_to_curate_GlotCC_are_not_validated_in_any_way._We_don't_know_how_the_resulting_corpus_compares_to_MADLAD,_CulturaX,_OSCAR,...
>
> We did not alter the text of any of the records in CommonCrawl, as done in previous works such as those from MADLAD and OSCAR. Both our LID and filters are designed to operate in the same manner: they either accept and label a record or reject it (or just label the error). Our focus is on minority languages, and as shown in Table 2, our effort covers a significantly larger number of languages compared to similar initiatives such as OSCAR. We will also provide summary statistics for the more major related corpora.
>
> As for language modeling, it would be pointless unless we also evaluate them, which requires evaluation data that we mostly do not have for minority languages (see [MEGA](https://aclanthology.org/2023.emnlp-main.258), [MEGAVERSE](https://aclanthology.org/2024.naacl-long.143/) and [Glot500](https://aclanthology.org/2023.acl-long.61/); for example, the [Belebele Dataset](https://aclanthology.org/2024.acl-long.44/) is one of the largest in terms of the number of languages supported for multiple-choice questions, covering 115 languages). At this stage, this venture seems unrealistic. We believe that identifying relevant data and performing language identification is a first step in that direction, but the actual training and evaluation of a new model would correspond to a separate project.
>
>
> > The_accuracy_of_GlotLID_is_not_measured_on_documents_selected_in_GlotCC,_except_in_Table_3._...
>
>
> We also conducted a sample audit on 653 languages, detailed in Table 8 of the supplementary material, which includes the in-language percentage and notes from the sample audit. We will expand the experiment in Table 3 to include more samples.
>
> > Section_3.4:_The_proposed_pipeline_does_not_perform_deduplication_due_to_its_computational_cost._However,_the_pipeline_could_have_taken_advantage_of_more_efficient_deduplication_pipeline,_such_as_...
>
> There are many deduplication methods with different parameters, which yield different results. We cannot measure the impact on language modeling, as we do not have evaluation data for minority languages. Also some methods might be advantageous for high-resource languages but not for minority languages. We would prefer to leave this decision to different user applications. We will perform a sample deduplication and report statistics on the estimated amount of duplicate data for each language.

---

### Official Review · Reviewer_xLgb · 2024-07-25
**A large-scale corpus covering 1000 languages**

**Rating:** 7
**Confidence:** 4
**Correctness:** The claims are most correct.
**Clarity:** the paper is well written.

**Review:**

* The paper is mostly well written with decent details on the dataset construction
* The released code and dataset should be interesting to the community for both low-resource study and multilingual modeling

**Strengths:**

* Extending the language identification model to GlotLID v3, which performs better and covers more languages
* Released GlotCC with 1000 languages; this would be a valuable resources for follow-up studies
* Open-source code to the community

**Additional Feedback:**

Please refer to Opportunities For Improvement section.

**Documentation:**

The authors released the source code for reproducing the data.

**Ethics:**

I didn't see serious issues.

**Limitations:**

The authors discussed limitations adequately.

**Opportunities For Improvement:**

* Consider adding results of previous models to Table 1.
* It would be great to train some models on this dataset and compare its performance with the others so as to better justify the significance of GlotCC.

**Relation To Prior Work:**

The relation to previous studies is properly discussed.

**Summary And Contributions:**

Large language model is data-hungry, which often performs worse on low-resource languages. A way to overcome this problem is to collect more data for these languages. In this study, the authors shared their recipe for the collection of a large-scale corpus, GlotCC, covering more than 1000 languages. They released the data and open sourced the creation pipeline to facilitate the research.

---

> ### Author Rebuttal · Authors · 2024-08-17
>
> Thank you for your positive feedback on the work! We are glad that you found the paper well written with decent details, and we answered the points raised in the review below:
>
>
> **Re: Consider adding results of previous models to Table 1**: We will include the results from https://arxiv.org/abs/2310.16248 for previous GlotLID versions in Table 1 of the paper for a better pairwise comparison.
>
> **Re: Training some models and compare with others**: Our focus is on minority languages, and as shown in Table 2, our effort covers a significantly larger number of languages compared to similar initiatives such as OSCAR. As for training some models, it would be pointless unless we also evaluate them, which requires evaluation data that we mostly do not have for minority languages (see [MEGA](https://aclanthology.org/2023.emnlp-main.258), [MEGAVERSE](https://aclanthology.org/2024.naacl-long.143/) and [Glot500](https://aclanthology.org/2023.acl-long.61/); for example, the [Belebele Dataset](https://aclanthology.org/2024.acl-long.44/) is one of the largest in terms of the number of languages supported for multiple-choice questions, covering 115 languages). At this stage, this venture seems unrealistic. We believe that identifying relevant data and performing language identification is a first step in that direction, but the actual training and evaluation of a new model would correspond to a separate project.

---

### Official Review · Reviewer_uMdT · 2024-08-02
**Need more experiments.**

**Rating:** 6
**Confidence:** 4
**Correctness:** Yes
**Clarity:** Yes, I think so.

**Review:**

The GlotCC presents a commendable effort in providing a diverse and trustworthy corpus for minority languages. While the work showcases strengths in terms of quality, clarity, originality, and significance, addressing potential biases, enhancing validation processes, clarifying technical jargon, benchmarking against existing corpora, and discussing broader impacts could further strengthen the work's overall contribution to the field of NLP and linguistic research.

**Strengths:**

The strengths of the submission lie in its significant contribution to linguistic diversity, relevance to the research community through a unique dataset and open-source pipeline, high-quality research methodologies, and the ethical and social considerations embedded in the project. These strengths position the submission as a valuable and impactful contribution to the field of NLP and language research.

**Additional Feedback:**

It lacks experiments about training LLMs with GlotCC. For example, the authors can try continue-training Llama with partitions of GlotCC and perform ablation studies.

**Documentation:**

Yes.

**Limitations:**

Yes

**Opportunities For Improvement:**

It lacks experiments about training LLMs with GlotCC. For example, the authors can try continue-training Llama with partitions of GlotCC.

**Relation To Prior Work:**

Yes.

**Summary And Contributions:**

The submission shows a significant contribution in the realm of text corpora for minority languages.

## Contributions:
1. Large Text Corpus Requirement: The paper addresses the growing necessity for extensive text corpora, especially in light of pretrained language models and the associated scaling laws.
2. Coverage of Minority Languages: GlotCC fills a crucial gap by providing a document-level, 2TB general domain corpus that spans over 1000 languages, including many minority languages often overlooked by existing corpora.
3. Trustworthy and Reproducible: The corpus is generated through an open-source and reproducible pipeline, ensuring transparency and reliability. Additionally, GlotCC undergoes rigorous cleaning processes to eliminate noise, enhancing its usability and trustworthiness for research purposes.

---

> ### Author Rebuttal · Authors · 2024-08-17
>
> Thank you for your feedback and for recognizing the requirement for GlotCC, as well as the emphasis on the trustworthiness and reproducibility of our method. We have answered the concerns raised in the review below:
>
>
> **Re: Clarifying Technical Jargon**: We made an effort to minimize technical jargon to ensure the content is understandable by a broader audience.
> We will proofread the text again to further clarify any remaining issues. If the reviewer has any specific sections in mind that need more attention, we would be happy to address them.
>
> **Re: Training LLMs with GlotCC and Benchmarking Against Existing Corpora**: We compare our work with OSCAR in terms of the number of languages and documents we support. Our focus is on minority languages, and as shown in Table 2, our effort covers a significantly larger number of languages compared to similar initiatives such as OSCAR. As for training LLMs, it would be pointless unless we also evaluate them, which requires evaluation data that we mostly do not have for minority languages (see [MEGA](https://aclanthology.org/2023.emnlp-main.258), [MEGAVERSE](https://aclanthology.org/2024.naacl-long.143/) and [Glot500](https://aclanthology.org/2023.acl-long.61/); for example, the [Belebele Dataset](https://aclanthology.org/2024.acl-long.44/) is one of the largest in terms of the number of languages supported for multiple-choice questions, covering 115 languages). At this stage, this venture seems unrealistic. We believe that identifying relevant data and performing language identification is a first step in that direction, but the actual training and evaluation of a new model would correspond to a separate project.
>
>
> **Re: Discussing Broader Impacts and Addressing Potential Biases**: We elaborated on the impact of data for more languages as we develop more NLP tools in the ethics statement section 7, and we will add more on the social impacts. Regarding potential biases, we discussed them in ethics statement section 7. Despite our strong focus on filtering, the cleaning of GlotCC is constrained by the lack of tools for filtering out undesirable content and noise—such as adult content, personal information, and lists—which we mostly do not have for minority languages. We believe that identifying relevant data and performing language identification is, again, the first step in that direction.

---

### Author Rebuttal · Authors · 2024-08-17

We thank the reviewers for their positive reviews and helpful comments regarding the **corpus and pipeline requirements** (uMdT: “GlotCC fills a crucial gap by providing a document-level, ... spans over 1000 languages, including many minority languages often overlooked by existing corpora.”), **language identification design and coverage** (6BqC: “The language identification approach presented in this work is well thought out ... this work puts great care in increasing coverage of languages that can be identified on web text”), **details and community interest** (xLgb: “The paper is mostly well written with decent details on the dataset construction .... the released code and dataset should be interesting to the community for both low-resource study and multilingual modeling”), **computational efficiency** (6BqC: “The proposed approach increases coverage of languages that can be identified without decreasing computational efficiency.”), and **trustworthiness and reproducibility** (uMdT: “The corpus is generated through an open-source and reproducible pipeline, ensuring transparency and reliability. Additionally, GlotCC undergoes rigorous cleaning processes to eliminate noise, enhancing its usability and trustworthiness for research purposes.”).

---

### Decision · Program_Chairs · 2024-09-26

**Decision:**

Accept (Poster)

**Comment:**

The paper introduces a pipeline for extracting multilingual text from CommonCrawl and provides a document-level corpus including over 1000 languages. All reviewers gave positive feedbacks on the contribution of providing large number of minority languages which may often be overlooked by existing corpora. One main issue also raised by ALL reviewers is the evaluation. Although large text corpora of minority languages is present, justification of its usefulness, at least preliminarily, is still important. In the rebuttal, the authors claimed that due to lack of testset, training a language model on the provided corpora is useless. However, there might be other ways, such as select a subset of minority languages for testing, to evaluate the provided corpora, especially against existing multi-lingual corpora (on the overlapped langauges). While the number of provided languages is certainly valuable, the quality/usefulness of the corpora is also of interest. I believe the "corpora evaluation" comments of ALL reviewers should still be carefully discussed in the paper and better be addressed using feasible experiments (could be small scale or partial).